# Activation of *Shigella flexneri* type 3 secretion requires a host-induced conformational change to the translocon pore

Brian C. Russo[1,2], Jeffrey K. Duncan[1¤a], Alexandra L. Wiscovitch[1,3¤b], Austin C. Hachey[1¤c], Marcia B. Goldberg[1,2]*

1 Center for Bacterial Pathogenesis, Department of Medicine, Division of Infectious Diseases, Massachusetts General Hospital, Boston, Massachusetts, United States of America, 2 Department of Microbiology, Blavatnik Institute, Harvard Medical School, Boston, Massachusetts, United States of America, 3 Research Scholar Initiative, The Graduate School of Arts and Sciences, Harvard University, Cambridge, Massachusetts, United States of America

¤a Current address: Yale School of Medicine, New Haven, Connecticut, United States of America
¤b Current address: Department of Molecular Genetics and Microbiology, University of Florida, Gainesville, Florida, United States of America
¤c Current address: Department of Chemistry, College of Arts and Sciences, University of Kentucky, Lexington, Kentucky, United States of America
* marcia.goldberg@mgh.harvard.edu

**Data Availability Statement:** All relevant data are within the manuscript and its Supporting Information files.

## Abstract

Type 3 secretion systems (T3SSs) are conserved bacterial nanomachines that inject virulence proteins (effectors) into eukaryotic cells during infection. Due to their ability to inject heterologous proteins into human cells, these systems are being developed as therapeutic delivery devices. The T3SS assembles a translocon pore in the plasma membrane and then docks onto the pore. Docking activates effector secretion through the pore and into the host cytosol. Here, using *Shigella flexneri*, a model pathogen for the study of type 3 secretion, we determined the molecular mechanisms by which host intermediate filaments trigger docking and enable effector secretion. We show that the interaction of intermediate filaments with the translocon pore protein IpaC changed the pore's conformation in a manner that was required for docking. Intermediate filaments repositioned residues of the *Shigella* pore protein IpaC that are located on the surface of the pore and in the pore channel. Restricting these conformational changes blocked docking in an intermediate filament-dependent manner. These data demonstrate that a host-induced conformational change to the pore enables T3SS docking and effector secretion, providing new mechanistic insight into the regulation of type 3 secretion.

## Author summary

The movement of bacterial proteins across membranes is essential for bacterial physiology and bacterial virulence. The type 3 secretion system moves bacterial virulence proteins from the inside of bacterial pathogens into human cells. To do so, the type 3 secretion system forms a pore in the plasma membrane of the target cell, attaches (docks) onto the

**Funding:** This work was funded by NIH RO1 AI081724 (M.B.G.), NIH T32 AI007061 (B.C.R.), NIH F32 AI114162 (B.C.R.) (https://www.niaid.nih.gov/), The Massachusetts General Hospital Executive Committee on Research Tosteson Award (B.C.R.) (https://ecor.mgh.harvard.edu/Default.aspx?node_id=316), and the Charles A. King Trust Postdoctoral Research Fellowship Program, Bank of America, N.A., Co-trustees (B.C.R.) (https://hria.org/tmf/KingBasic). The funders had no role in study design, data collection and analysis, decision to publish, or preparation of the manuscript.

**Competing interests:** The authors have declared that no competing interests exist.

pore, and delivers virulence proteins through the pore. Docking is essential for establishing a continuous channel from the inside of the bacterium to the inside of the human cell. What enables the type 3 secretion system to dock onto pores is not understood. We show that structural proteins in human cells, intermediate filaments, induce structural rearrangements to the type 3 secretion pore that trigger docking and that enable the subsequent delivery of virulence proteins into human cells. Due to the widespread prevalence of type 3 secretion systems among human pathogens, these findings are likely to broadly enhance our understanding of type 3 secretion.

## Introduction

Type 3 secretion systems (T3SSs) are present in and are essential for the virulence of more than 30 bacterial pathogens of humans, animals, and plants. T3SSs deliver bacterial effector proteins into the cytosol of eukaryotic host cells. In the eukaryotic cytosol, effector proteins manipulate cellular signaling in ways that promote bacterial virulence. The T3SS apparatus consists of a base that spans both bacterial membranes, a needle that is anchored in the base and extends from the bacterial surface, and a tip complex that is positioned at the distal end of the needle and prevents non-specific secretion [1–3]. Whereas effector proteins are specific to each pathogen, the T3SS apparatus is highly conserved across bacterial species [4].

The molecular context that triggers secretion and translocation of effector proteins through the T3SS apparatus into the host cell is complex. Contact with the target cell induces the T3SS to form the translocon pore in the plasma membrane [5]; this translocon pore is a heterooligomer of two bacterial proteins secreted through the T3SS [3, 6]. The *Shigella* translocon pore protein IpaC and its homologs interact with intermediate filaments [7–9], eukaryotic cytoskeletal proteins. For *Shigella*, this interaction is not required for the formation of the pore; rather, this interaction is required for docking (or attachment) of the T3SS needle onto the pore [7]. Docking is necessary for the secretion of effector proteins through the T3SS [7, 10], making this step a critical checkpoint for effector secretion. Neither the signal that activates secretion nor the mechanism by which docking is enabled by the interaction of intermediate filaments with the pore has been determined.

Contact with the host cell is required to activate the T3SS, but how host contact is sensed at the site of secretion activation in the bacterial cytosol, at a distance of ~100 nm from the site of host contact, is unknown. By comparing the structures of T3SS that are competent for secretion to those that are not, conformational changes to the T3SS needle and base have been observed in some but not all studies [11–13]. Additional evidence demonstrates that the translocon pore, rather than simply a conduit for effector secretion, functions in an active capacity to regulate effector secretion [7, 10]. Discrete mutations in translocon pore proteins have been identified that do not alter the efficiency of translocon pore formation but restrict the efficiency of docking or effector translocation [7, 14], which shows that translocon pore formation *per se* is not sufficient for T3SS function.

We and others hypothesize that host contact generates a series of conformational changes along the T3SS needle that constitute a signal from the extracellular environment into the bacterial cytoplasm [7, 15–17], where effector secretion is orchestrated [12, 18, 19]. Because the T3SS translocon pore is situated in the host membrane [6, 8, 20–22], its interactions with the host could be the initial sensor that instigates this signaling cascade [10, 16, 23]; however, how the translocon pore would generate such a signal and specifically whether conformational

changes that might contribute to such a signal occur in the translocon pore and/or needle are unknown.

For several pathogens that utilize type 3 secretion, host intermediate filaments interact with translocon pore proteins [7–9]. For *S. flexneri*, intermediate filaments are required for efficient docking [7]. Since docking is necessary for activation of *S. flexneri* effector secretion [7] and effector protein secretion requires a conformational change to an orthologous translocon pore [10], we hypothesized that the interaction of translocon pore proteins with intermediate filaments specifically induces conformational changes required for docking and subsequent docking-dependent activation of type 3 secretion. To test this hypothesis, we compared the positioning and accessibility to the extracellular milieu of residues across the length of the *S. flexneri* translocon pore protein IpaC following its native delivery to the plasma membrane under conditions that support or restrict bacterial docking. We demonstrate that an intermediate filament-induced conformational change to the translocon pore is required for docking. We propose that activation of secretion is induced by a signaling relay into the bacterial cytoplasm that is initiated by this host-induced conformational change in the translocon pore.

## Results

### In membrane-embedded translocon pores, identification of residues of IpaC that are amenable to intermolecular crosslinking

Since the interaction of intermediate filaments with the *Shigella* translocon pore protein IpaC is dispensable for pore formation but required for docking [7], we hypothesized that the interaction of IpaC with intermediate filaments alters the conformation of the translocon pore in a manner that enables docking. To determine the impact of intermediate filaments on T3SS activity, we first sought to identify residues of IpaC at which intermolecular crosslinks could be established, as we predicted that crosslinks would stabilize particular conformations of the pore, as previously described for the *Pseudomonas aeruginosa* T3SS [10].

We took advantage of the ability of the oxidant copper to induce disulfide crosslinks between cysteine residues that reside in an oxidative environment in close proximity to one another [10, 24]. We screened a library of functional IpaC cysteine substitution mutants [23] for residues that within the pore lie sufficiently close to one another to form disulfide bonds; wild-type (WT) IpaC lacks cysteines. Because we hypothesized that the interaction of IpaC with intermediate filaments induces a conformational change in the translocon pore, we looked for residues of IpaC for which intermolecular proximity was maintained in the presence and in the absence of intermediate filaments, as these residues would permit analysis of the impact of disulfide crosslinking on IpaC function.

As *S. flexneri* infections were performed in the presence of copper, disulfide bonds likely form as soon as the pore multimerizes. Pores are more likely to form in the extracellular environment, which is oxidizing, than in the cytosolic environment, which is reducing (Fig 1a); the oxidation state of the pore channel is unknown. Thus, we anticipated formation of intermolecular disulfide bonds in the subset of IpaC cysteine substitutions that are extracellular, or potentially in the pore channel, and that are oriented correctly and situated in close proximity to one another. The overall topology of IpaC, which we defined previously by testing the accessibility of residues to a membrane-impermeant 5,000-kilodalton probe (PEG5000) from the extracellular space, is N-terminal domain extracellular and C-terminal domain cytosolic [23]. Here, we complemented that approach by identifying IpaC residues that are simultaneously accessible to the much smaller membrane-impermeant probe copper chloride (58 daltons) and positioned such that they could be crosslinked. Seven IpaC cysteine substitutions, S17C, A38C, S63C, A106C, K350C, A353C, and A363C, displayed a band that migrated at ~80 kilodaltons,

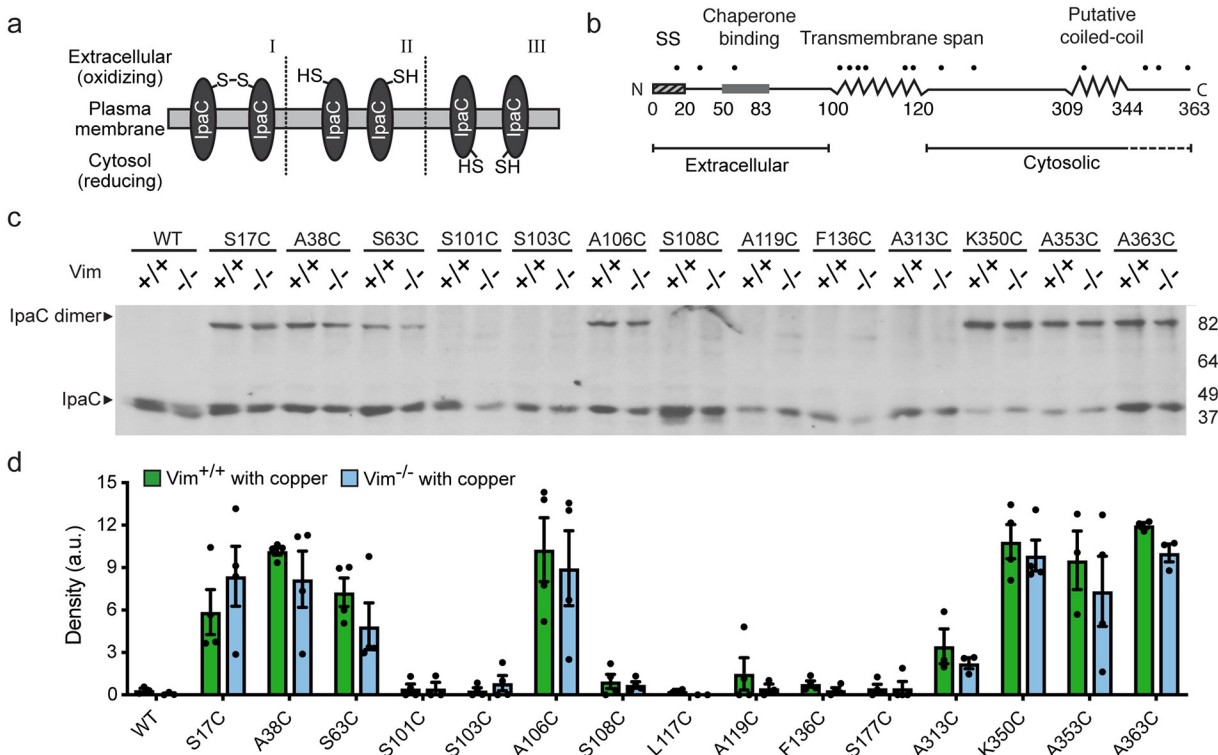

**Fig 1. In membrane-embedded translocon pores, individual IpaC molecules are adjacent to one another. (a)** Schematic of the dependence of disulfide bond formation on an oxidative environment and the relative positioning of free sulfhydryl groups. Disulfide bond formation occurs when residues are located in the extracellular space, and when in close proximity with the correct orientation of the residues in adjacent molecules [I]. The formation of disulfide bonds is not efficient when residues are in an oxidative environment but are not in proximity or correctly oriented [II], or when residues are in a reducing environment, such as the cytosol [III]. **(b)** Schematic of secondary structure and orientation in the membrane of IpaC protein. Dots indicate the relative position of the mutants tested. Dashed line beneath residues 344–363 reflects the accessibility of these residues to the extracellular milieu [23], suggesting that they loop into the pore lumen. **(c-d)** Analysis of intermolecular crosslinking of single cysteine substitutions of IpaC delivered to cells during *S. flexneri* infection. Infection of Vim+/+ or Vim-/- MEFs in the presence of the oxidant copper with strains of *S. flexneri* ΔipaC producing individual IpaC cysteine substitutions. Representative western blot of IpaC, showing bands migrating at the molecular weight of IpaC dimers and monomers (**c**). Densitometry of the ~80 kilodalton band from three to four independent experiments; mean ± SEM (**d**). For cysteine substitution derivatives, band densities are not significantly different for infection of Vim+/+ versus Vim-/- MEFs.

more slowly than WT IpaC (Fig 1c and 1d), consistent with the presence of a disulfide bond between adjacent IpaC molecules and consistent with our previous demonstration that each of these residues is accessible from the extracellular surface of host cells [23]: IpaC S17, A38, and S63 are within the extracellular N-terminal domain, A106 is within the single transmembrane span that contributes to the pore channel, and K350, A353, and A363 are within a stretch of residues near the C terminus that appears to loop back into the pore channel (Fig 1b). In contrast, we did not observe the formation of disulfide bonds at S101C, S103C, S108C, L117C, A119C, F136C, S177C, or A313C, which lie in the transmembrane span (S101C, S103C, S108C, L117C, and A119C) and in the cytosol (F136C, S177C, and A313C) [23].

The residues at which disulfide bonds formed and the efficiency with which they formed were not different for mouse embryonic fibroblasts (MEFs) that contain vimentin (Vim+/+) or lack vimentin (Vim-/-), which indicates that either the translocon pores that form upon initial contact with cells are in similar conformational states in the presence and absence of the IpaC-intermediate filament interaction or that, at these residues, adjacent IpaC molecules remain close despite any potential conformational difference between the two states.

It was plausible that copper could have a non-specific effect on the formation of translocon pores. We therefore tested the impact of copper on the efficiency of translocon pore formation in the plasma membranes of erythrocytes. In this assay, formation of a translocon pore causes erythrocyte lysis and hemoglobin release, such that hemoglobin abundance in the media is a measure of the efficiency of translocon pore formation [6, 7, 25]. Hemoglobin released from erythrocytes co-cultured with strains of *S. flexneri* producing WT IpaC, IpaC A38C, or IpaC A353C was unchanged by the presence of copper (S1a and S1b Fig). Copper did not cause erythrocyte lysis on its own, nor did it change the dependence of hemoglobin release on IpaC (S1a and S1b Fig). These data demonstrate that neither the formation of intermolecular bonds between IpaC molecules nor the presence of copper *per se* interferes with pore formation.

To further assess whether the intermolecular bonds that formed between these IpaC residues were accessible from the extracellular side of the host cell, we carried out copper-induced crosslinking in the presence of the membrane-impermeant reductant tris-(2-carboxyethyl) phosphine hydrochloride (TCEP). Addition of TCEP markedly and significantly reduced disulfide bond formation at all residues except A363C, and at A363C, it reduced disulfide bond formation by 50%, which did not achieve statistical significance (S1c and S1d Fig). Since both copper chloride and TCEP are membrane impermeable, the formation of disulfide bonds at S17C, A38C, S63C, A106C, K350C, A353C, and some of the bonds formed at A363C, are most likely accessible from the extracellular environment. Moreover, the observation that crosslinking of A38C and A353C results in a defect in docking (see below) indicates that, for at least these two residues, crosslinking involves monomers that are within translocon pores. Together, these data show that the positioning of individual IpaC molecules within the pore enables the formation of intermolecular disulfide bonds.

### Intermolecular crosslinking of IpaC inhibits intermediate filament-dependent docking of the type 3 secretion system onto the translocon pore

We hypothesized that if interactions of the translocon pore with intermediate filaments alter the pore conformation in a manner that enables docking, crosslinking the pore might restrict movement of the pore proteins, thereby trapping it in a docking-incompetent state, leading to fewer bacteria stably associated with cells. To test this, we examined bacterial docking in Vim$^{+/+}$ and Vim$^{-/-}$ cells infected with *S. flexneri* expressing individual cysteine substitution derivatives or WT IpaC in the presence or absence of copper. We found that copper significantly reduced the efficiency of docking of *S. flexneri* producing IpaC A38C or IpaC A353C to Vim$^{+/+}$ cells (Fig 2a and 2b). In the absence of copper, docking to Vim$^{+/+}$ cells is similar for these strains and a strain of *S. flexneri* producing WT IpaC (Fig 2a and 2b), which indicates that the observed reduction in docking is not due to the cysteine substitutions *per se*, in agreement with our previous data that demonstrate that these cysteine substitution derivatives of IpaC are functional [23]. Moreover, since the efficiency of docking of *S. flexneri* producing WT IpaC to Vim$^{+/+}$ cells was unaltered by the presence of copper (Fig 2a and 2b), the impact of copper on bacterial docking is specifically due to formation of a disulfide bond at the sulfhydryl group incorporated into IpaC at A38 or at A353.

As we reported previously [7], docking of *S. flexneri* to Vim$^{-/-}$ cells is 5-fold less efficient than docking of *S. flexneri* to Vim$^{+/+}$ cells (Fig 2a and 2b). The formation of intermolecular disulfide bonds *per se* does not inhibit docking to Vim$^{-/-}$ cells (Fig 2b), indicating that copper-induced crosslinking inhibits the docking process by blocking an intermediate filament-dependent shifting of IpaC in the pore. However, because the levels of docking to Vim$^{-/-}$ cells are close to and possibly at background levels, from these data alone, we cannot definitively determine whether or not the disulfide crosslink also interferes with docking independent of

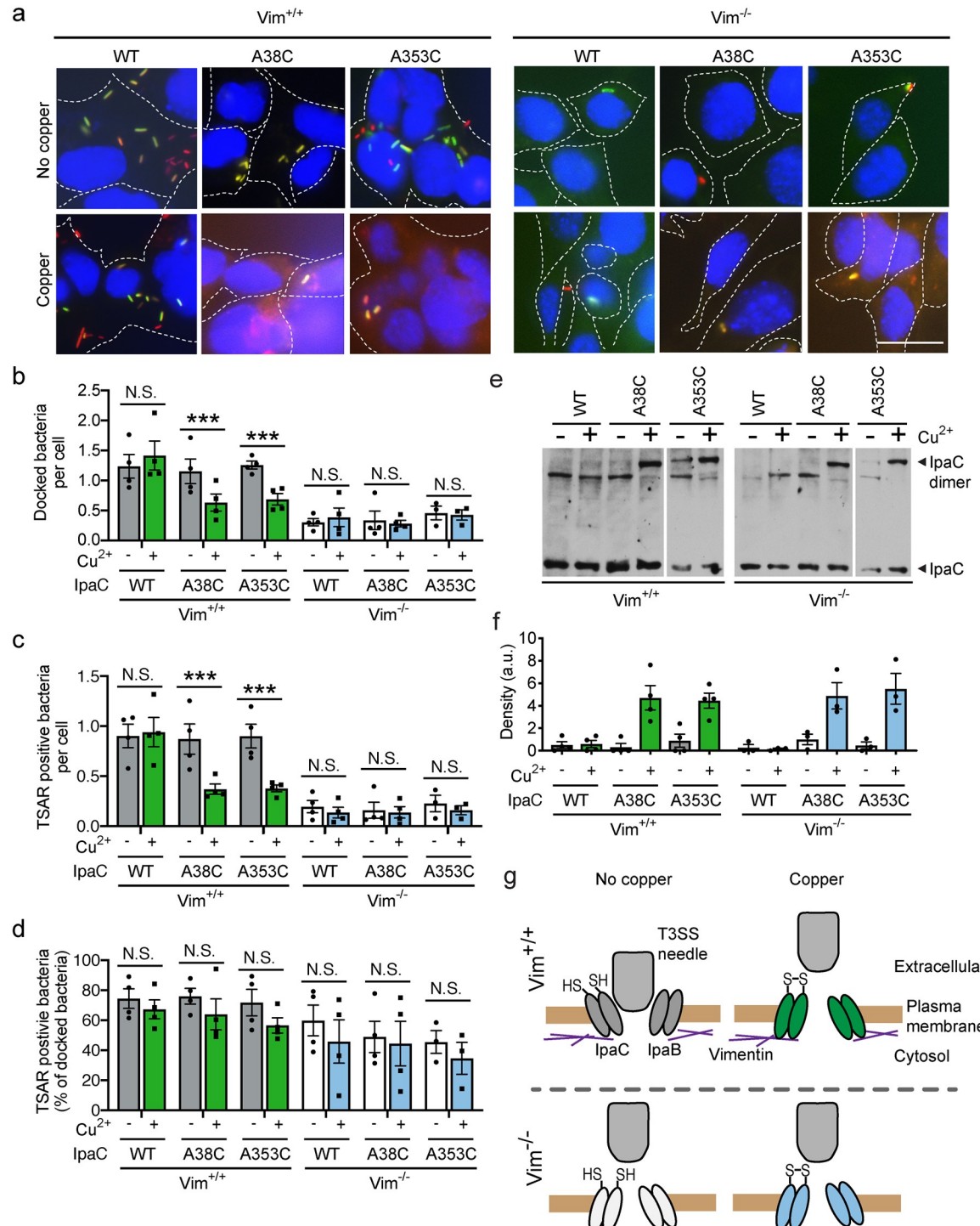

**Fig 2. Intermolecular crosslinking of IpaC molecules in the translocon pore at the time of initial bacterial contact with cells inhibits docking.** Efficiency of docking and T3SS-mediated secretion upon addition of the oxidant copper during *S. flexneri* infection of Vim$^{+/+}$ or Vim$^{-/-}$ MEFs. Copper was added at the time of initial bacterial contact with cells. IpaC or its derivatives were expressed in *S. flexneri ΔipaC* containing the TSAR reporter of T3SS secretion [37]. **(a)** Representative micrographs of *S. flexneri*-infected cells, imaged at 50 minutes of infection. Blue, DNA; red, mCherry (all bacteria); green, GFP (bacteria actively secreting through T3SS). Scale bar, 20 μM. **(b)** Efficiency of docking of bacteria on cells. **(c-d)** The number of docked bacteria per cell (**c**) and the percentage of docked bacteria that are actively secreting (**d**), as indicated by TSAR reporter, from images in experiments represented in panel **a**. **(e)** IpaC dimer formation analyzed at 20 minutes of infection. Non-reducing western blot of IpaC, representative of three independent experiments; all panels are from the same blot. **(f)** Densitometry analysis of bands corresponding to IpaC dimers in experiments

represented in panel **e**. **(g)** Model for the effect of IpaC crosslinking on restricting shifting of the translocon pore in the plasma membrane. Graphed data are presented as mean ± SEM of three to four independent experiments. *, p<0.05; **, p<0.01; ***, p<0.001; two-way ANOVA with Sidak's *post hoc* test (**b-d**).

intermediate filaments. In either case, copper-induced crosslinking restricted an intermediate filament-dependent process that is required for *S. flexneri* docking.

As expected, the inhibition of docking of bacteria to Vim$^{+/+}$ cells was associated with a concomitant reduction of the number of bacteria that activated T3SS secretion (Fig 2a and 2c). Yet, among the small subset of bacteria that successfully docked, the presence of copper did not significantly impact the efficiency of T3SS-mediated secretion (Fig 2a and 2d), which suggests that crosslinking interrupts docking but not docking-induced activation of secretion.

Although the restriction of docking by copper-induced crosslinks was specific for translocon pores formed in the presence of vimentin (Fig 2a and 2b), intermolecular crosslinking of IpaC was equally efficient in Vim$^{+/+}$ and Vim$^{-/-}$ cells (Fig 2e and 2f). Moreover, the amount of IpaC monomer delivered to Vim$^{-/-}$ and Vim$^{+/+}$ cells was unchanged by the presence of copper (Fig 2e). Together, these data demonstrate that crosslinking the translocon pore specifically interrupts intermediate filament-dependent shifting of the pore required for efficient docking (Fig 2g).

## Interaction of IpaC with intermediate filaments induces conformational changes in the translocon pore

We hypothesized that conformational changes induced by the interaction of IpaC with intermediate filaments might be detected as altered accessibility of residues of the translocon pore proteins to the extracellular surface. To test this hypothesis, we compared the extracellular accessibility of single cysteine substitutions in IpaC in the presence and absence of interaction of IpaC with intermediate filaments. IpaC cysteine substitutions were labeled by methoxypolyethylene glycol maleimide (PEG5000-maleimde), a probe that reacts specifically with sulfhydryl groups present in cysteines, as described [23]. Because PEG5000-maleimide is unable to cross the plasma membrane [26] and is too large to pass through the translocon pore [6, 7], this approach specifically labels cysteine residues accessible from the extracellular surface of the eukaryotic cell (Fig 3a).

The extracellular accessibility of individual cysteine substitutions in IpaC was compared to those in a derivative of IpaC that contains a point mutation (IpaC R362W [27], Fig 3b) that prevents interaction of IpaC with intermediate filaments but not pore formation [7]. First, to test whether the overall orientation of IpaC R362W in the membrane is similar to that of WT IpaC, HeLa cells were infected with strains of *S. flexneri* producing IpaC R362W containing single cysteine substitution mutations. HeLa cells express multiple intermediate filaments that are present in cells of the intestine, including vimentin, keratin 8, and keratin 18 [7]. PEG5000-maleimide was added at the start of infection [23]. Plasma membrane-enriched fractions were isolated, and the efficiency by which PEG5000-maleimide labeled IpaC was determined by the rate of migration of IpaC through an SDS-PAGE gel. A slower migrating band indicative of PEG5000-labeled IpaC [7] was more abundant for substitutions near the N-terminus than for substitutions C-terminal to the IpaC transmembrane span, which indicates that the N-terminal region of IpaC R362W was present on the extracellular surface of the host cell membrane and the C-terminal two-thirds of IpaC R362W is primarily located on the cytosolic side of the membrane (S2a and S2b Fig). The overall orientation in the membrane of IpaC R362W, with the N-terminal region extracellular and the C-terminal region cytosolic, is the same as that of WT IpaC [23].

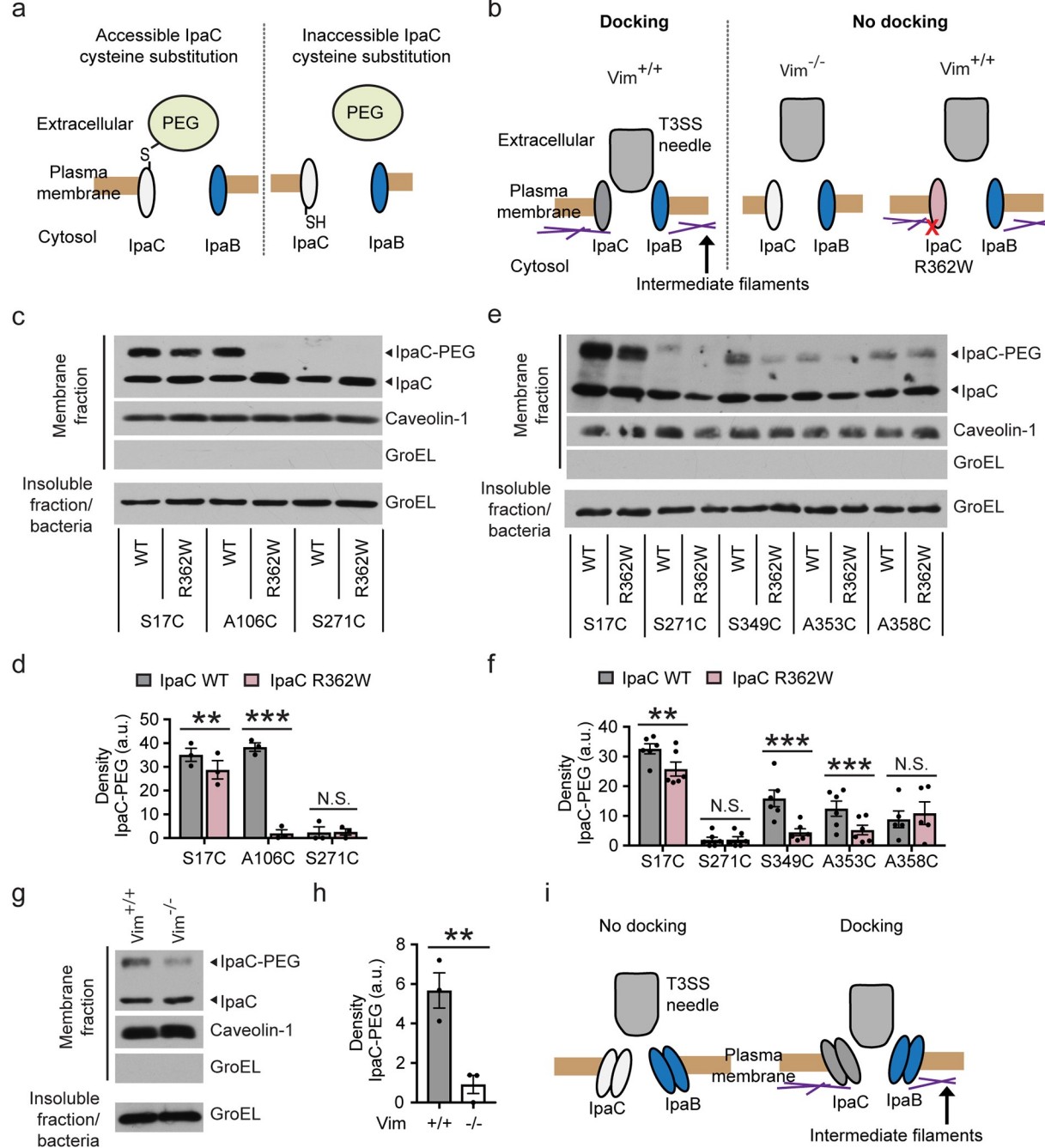

**Fig 3. Accessibility of IpaC residues from the extracellular surface is altered by the interaction of IpaC with intermediate filaments. (a)** Schematic of PEG5000-maleimide labeling of IpaC residues accessible from the extracellular face of the plasma membrane. PEG, PEG5000-maleimide. **(b)** Schematic of docking phenotype for translocon pores formed by WT IpaC or IpaC R362W, a derivative of IpaC unable to interact with intermediate filaments. **(c-f)** PEG5000-maleimide labeling of sulfhydryl groups in cysteine substitution derivatives in the context of WT IpaC or IpaC R362W during *S. flexneri* infection of HeLa cells. PEG5000-maleimide labeling of residue A106C in the transmembrane span (**c-d**) and residues S349C and A358C near the C terminus (**e-f**). Representative western blot of IpaC in plasma membrane-enriched fractions following PEG5000-maleimide labeling during infection (**c** and **e**). Densitometry analysis of bands corresponding to IpaC-PEG5000 (**d** and **f**). IpaC-PEG5000, IpaC derivatives labeled with PEG5000-maleimide; IpaC, unlabeled IpaC derivatives. **(g-h)** Decreased accessibility of IpaC A106C during infection of cells lacking intermediate filaments (Vim$^{-/-}$) as compared with cells containing intermediate filaments (Vim$^{+/+}$). Representative western blot of IpaC in plasma membrane-enriched fractions following PEG5000-maleimide labeling during infection with *S. flexneri* producing IpaC A106C (**g**). Caveolin-1, marker of eukaryotic plasma membrane; GroEL, bacterial cytosolic protein. Densitometry analysis of IpaC-PEG5000 bands (**h**). **(i)** Model of the dependence of docking of the T3SS on conformational changes in the translocon pore induced by the interaction of IpaC with intermediate filaments. Graphed data are presented as mean ± SEM of three

independent experiments. Two-way ANOVA with a Sidak *post hoc* test (**d** and **f**) or Student's t-test (**h**). **, p<0.01; ***, p<0.001, N.S., not significant.

*S. flexneri* strains producing IpaC R362W cysteine substitutions induced hemoglobin release from erythrocytes as efficiently as WT IpaC (S2c and S2d Fig), which indicates that the cysteine substitutions do not alter the efficiency of translocon pore formation, consistent with single cysteine substitutions in the context of WT IpaC not altering pore formation (S1a and S1b Fig, and [23]) and our previous demonstration that IpaC R362W forms pores at similar efficiency to WT IpaC [7]. Moreover, the amount of IpaC delivered to the membrane of cells was not altered by the R362W mutation (S2e Fig). Thus, IpaC R362W is delivered to the membrane in the same overall orientation in the membrane as WT IpaC and forms translocon pores with similar efficiency to WT IpaC.

Although the orientation of IpaC R362W in the plasma membrane is similar to that of WT IpaC, in translocon pores containing IpaC R362W, several individual IpaC residues displayed markedly reduced labeling. To assess the position of these residues in docking-competent versus docking-incompetent pores, accessibility from the extracellular milieu of individual sulfhydryl groups was compared during infection of HeLa cells with *S. flexneri* producing analogous cysteine substitutions in the WT IpaC and in the IpaC R362W backbones. Labeling by PEG5000-maleimide of the transmembrane span residue A106C was significantly reduced for IpaC R362W compared to WT IpaC (Fig 3c and 3d). In addition, labeling of the N-terminal residue S17C was slightly but significantly reduced for IpaC R362W compared to WT IpaC, whereas labeling of the cytosolic residue S271C was absent for both IpaC R362W and WT IpaC (Fig 3c and 3d). These findings indicate that the position in the pore of IpaC residues A106 and S17 is substantially different between the WT IpaC backbone and the IpaC R362W backbone.

We also examined residues in the C-terminal domain that appeared to be differentially accessible in the presence and absence of an interaction between IpaC and intermediate filaments. We found that extracellular accessibility of S349C and A353C by PEG5000-maleimide was less efficient in docking-incompetent than in docking-competent pores (Fig 3e and 3f and S3 Fig). In contrast, A358C was similarly accessible in these two contexts (Fig 3e and 3f and S3 Fig). Again, there was a small but statistically significant reduction in the extracellular accessibility of S17C in IpaC R362W compared to S17C in WT IpaC, but no difference in the extracellular accessibility of S271 (Fig 3e and 3f and S3 Fig). Since the difference we observed in docking-competent pores was increased accessibility to PEG5000 maleimide for certain cysteines, this difference is not due to steric hindrance from docked bacteria. Together, these data show that the accessibility of some C-terminal IpaC residues from the cell surface is enhanced by the interaction of IpaC with intermediate filaments.

Since in pores that could not interact with intermediate filaments, A106C was inaccessible from the extracellular milieu, we tested whether the extracellular accessibility of A106C was also decreased in cells that lack intermediate filaments. We found that IpaC A106C was significantly less accessible in the absence than in the presence of vimentin (Vim$^{-/-}$ versus Vim$^{+/+}$) (Fig 3g and 3h). Whereas for IpaC R362W A106C we observed no labeling by PEG5000-maleimide, for WT IpaC A106C in Vim$^{-/-}$ cells, we observed a small amount of labeling; this may be due to differences between the cell types (HeLa versus MEFs), differences in the kinetics of the infection, or functional defects of IpaC R362W beyond a lack of interaction with intermediate filaments. Nevertheless, our results demonstrate that the decreased extracellular accessibility of A106C in the IpaC R362W backbone is not simply due to non-specific effects from the R362W substitution *per se*, but rather to the absence of the interaction of IpaC with intermediate

filaments. Altogether, these data indicate that interaction of IpaC with intermediate filaments induces a conformational change in the translocon pore (Fig 3i).

To test whether interaction with intermediate filaments also induces conformational changes in other type 3 secretion systems, we examined the type 3 secretion system of *Salmonella* Typhimurium, which is closely related to that of *S. flexneri*. Since the *S.* Typhimurium translocon pore protein SipC, a homolog of IpaC, is required for stable docking of *S.* Typhimurium onto cells [28], and an interaction of SipC with intermediate filaments is required for effector translocation [7], we tested whether the interaction of SipC with intermediate filaments induces a conformational change in SipC. We compared the accessibility of SipC residues from the extracellular surface in the presence and absence of intermediate filaments, focusing on residues of SipC that are homologous to IpaC residues that displayed intermediate filament-dependent changes in accessibility (Fig 3). Among the SipC residues tested, only S18C and S38C were accessible to PEG5000-maleimide, and the absence of intermediate filaments was not associated with accessibility of additional residues (S4a–S4d Fig). When the efficiency of labeling of SipC residues was directly compared for $Vim^{+/+}$ and $Vim^{-/-}$ cells, there was not a statistically significant difference at either S18C or A38C (S4e and S4f Fig). The restriction of SipC extracellular accessibility to a few residues in its N-terminal domain is a notable difference from accessibility of IpaC and is consistent with our previous findings on the accessibility of SipC residues to PEG5000-maleimide during infection of HeLa cells [23].

## Docking occurs within 10 minutes of contact with host cells

Since we observed that intermolecular crosslinking at the time of initial bacterial contact with cells locked the translocon pore in a docking-incompetent state (Fig 2), we postulated that the addition of copper at this early time might restrict the pore in an initial conformation that exists prior to conformational changes induced by intermediate filaments. To test this possibility, we assessed whether at a slightly later infection time copper-induced crosslinking inhibited bacterial docking. In contrast to the impact of adding copper at the time of initial bacterial contact with cells, the addition of copper at 10 minutes after initial bacterial contact had no effect on docking; in the presence of copper, docking and type 3 secretion were similar for the strain producing IpaC A38C and the strain producing WT IpaC (Fig 4a–4c). Whereas docking and type 3 secretion were not affected, the addition of copper at 10 minutes post-initiation of infection led to formation of intermolecular crosslinks in IpaC A38C (Fig 4d and 4e). Although theoretically possible, it is unlikely that the observed crosslinking was associated with pores formed by newly-docked bacteria, since the majority of the intracellular pool of IpaC and IpaB is secreted within 9 minutes of initial bacterial contact with cells [29]. Altogether, these data suggest both that the vast majority of translocon pores are formed in the 10-minute period prior to the addition of copper and that processes required for docking are complete within 10 minutes of initial bacterial contact with cells.

## Discussion

The translocon pore of T3SSs serves not only as the conduit through which bacterial effector proteins are translocated across the plasma membrane of host cells, but also as a platform for both docking of the T3SS needle onto the pore and signaling that the system is primed for effector secretion. Docking is essential for activation of secretion of effector proteins through the T3SS [10, 23]. We describe conformational changes in the *Shigella* translocon pore that transpire within minutes of the initial contact of bacteria with the host cell. These conformational changes are dependent upon the interaction of the pore protein IpaC with intermediate filaments. Although the interaction with intermediate filaments occurs in the host cytosol,

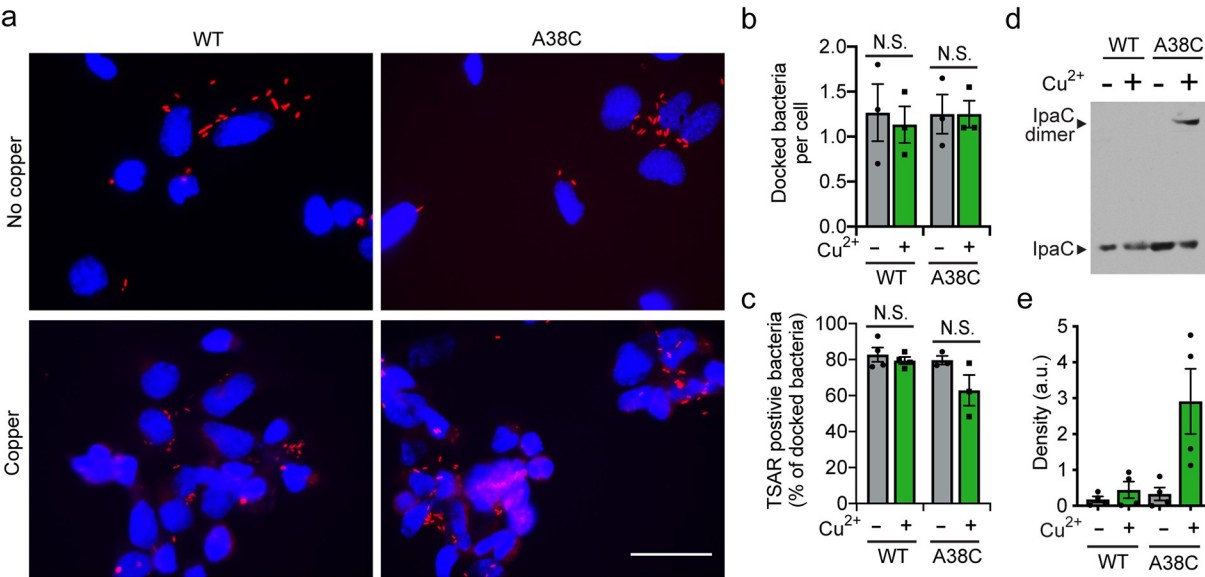

**Fig 4. A docking-competent conformation is established within 10 minutes of initial contact of *S. flexneri* with cells.** Induction of disulfide crosslinking of IpaC cysteine substitution derivatives by addition of the oxidant copper 10 minutes after initial bacterial contact with cells. **(a)** Representative micrographs of *S. flexneri*-infected cells, imaged at 50 minutes of infection, so as to allow production of GFP from TSAR. Blue, DNA; red, mCherry (bacteria). Scale bar, 50 μM. **(b)** Quantification of docked bacteria per cell, in experiments represented in panel **a**. **(c)** Percent of docked bacteria that activated secretion, as measured by the TSAR reporter. **(d)** Representative non-reducing western blot of IpaC in plasma membrane-enriched fractions, showing dimer formation by IpaC A38C in the presence of copper. **(e)** Densitometry analysis of bands corresponding to the IpaC dimer in experiments represented in panel **d**. Graphed data are presented as mean ± SEM of three or more independent experiments. N.S., not significant; **, $P<0.01$. Two-way ANOVA with Sidak *post hoc* test.

involving the cytosolic C-terminal region of IpaC [7], we provide evidence that the resulting conformational change includes shifts not only in sequences near the point of contact with intermediate filaments, but also in the channel of the translocon pore and in extracellular N-terminal sequences (Fig 5). IpaC residues S17 in the N-terminal extracellular domain, A106 in the transmembrane alpha helix, and S349 and A353 near the C-terminus display significant differences in extracellular accessibility between the docking-competent and the docking-incompetent states of the translocon pore (Fig 3). The dramatic change in accessibility of A106 suggests that the transmembrane alpha helix (residues 100–120) undergoes a rotation such that upon IpaC interaction with intermediate filaments, A106 turns out of the lipid bilayer into the pore channel or is relieved from steric hindrance by other sequences in IpaC or by the second pore protein IpaB. We envision that the rotation of the transmembrane alpha helix is induced by IpaC C-terminal binding to intermediate filaments and is accompanied by movement of residues near the C-terminus into the pore channel and shifting of the N-terminal domain, potentially in a manner that opens the extracellular side of the pore (Fig 5). We postulate that these conformational changes alter the pore so as to accommodate the T3SS needle, thereby leading to efficient docking. Our data are consistent with the previous observation in *P. aeruginosa* of the T3SS needle tip complex contacting residues in the C-terminal domain of PopD [10], the *P. aeruginosa* homolog of IpaC. If, as our model proposes, residues near the IpaC (or PopD) C-terminus loop into the pore lumen (Fig 3e and 3f and [23]), then during docking, if the needle tip complex partially enters the pore, it would be positioned to engage these residues (Fig 5).

We and others postulated that effector secretion is activated by a signal(s) communicated from the extracellular site of needle docking to the T3SS sorting platform, a dynamic protein

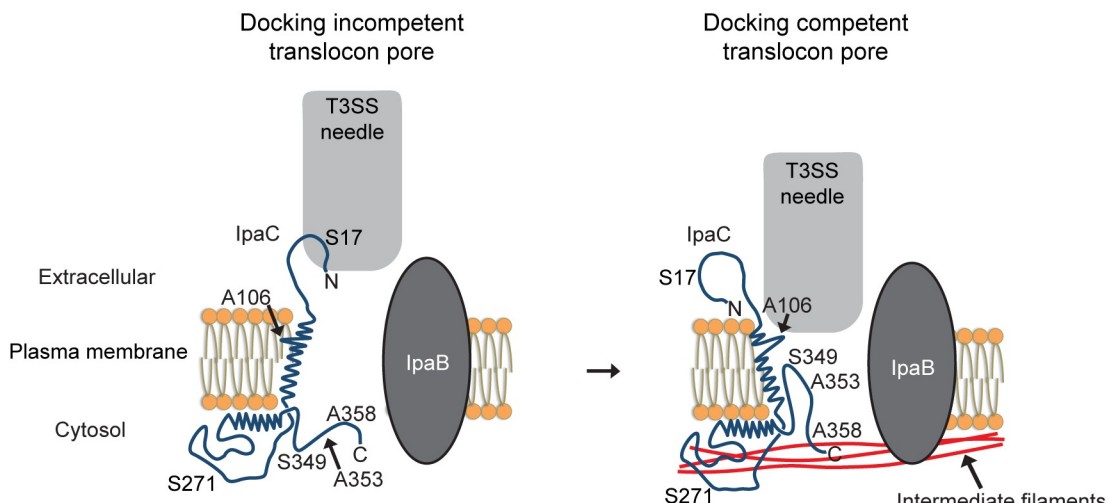

**Fig 5. Model of conformational changes induced by interaction of IpaC with intermediate filaments.** The translocon pore is formed in the plasma membrane such that the N-terminal region is extracellular and that the C-terminal region is cytosolic [23]. In the absence of an interaction between IpaC and intermediate filaments, residues A106, S349, and A353 are not readily accessible from the extracellular space. Interaction of IpaC with intermediate filaments is associated with significantly greater extracellular accessibility of A106, S349, A353, and S17C. In contrast, IpaC interaction with intermediate filaments does not alter the accessibility of A358 or S271.

complex situated in the bacterial cytoplasm at the base of the T3SS apparatus [3, 12, 15, 18, 30]. The sorting platform coordinates the secretion of effector proteins out of the bacterial cytoplasm and into the needle [12, 18]. A mounting body of evidence suggests that conformational changes in the T3SS needle and base are required to activate type 3 secretion [15, 16, 31, 32]. Imaging studies that are sufficiently sensitive to detect subtle changes in conformation of the T3SS needle and base show these structures are altered by host contact [11, 32]. We postulate that the observed conformational changes in the needle and base are directly triggered by docking and in turn activate the sorting complex to secrete effector proteins. The conformational changes we identify in the translocon pore represent the earliest molecular switch associated with activation of secretion and likely represent an initial trigger of a signaling cascade that directly regulates this process.

We previously showed that docking is required for activation of effector secretion [7], but it is uncertain whether docking and the associated conformational change we observe is sufficient to activate secretion or whether additional conformational changes and/or signals are also required. A number of secretion inducing signals have been identified, including the recognition of intracellular $Ca^{2+}$ [31], actin polymerization via Src recruitment [33], intracellular pH [34], lipid rafts [35, 36], and amino acids [31]. For *Shigella*, actin polymerization is associated with an IpaC-dependent activation of effector secretion [33, 37], but the role of actin polymerization in *S. flexneri* docking and effector secretion is not understood in molecular detail. The region of IpaC near the C-terminus is important both for the recruitment of Src [33] and the nucleation of actin filaments [38]. The particular residues of IpaC required for these processes are unknown; single residue scanning mutagenesis of the C-terminal region of IpaC including R362 did not identify mutants that blocked actin nucleation [38]. An insertion of the C3 epitope at A351 of IpaC disrupted Src recruitment to the site of *S. flexneri* invasion, but whether this insertion disrupts docking was not tested [33]. Interestingly, the regions of IpaC that are required to recruit Src and nucleate actin polymerization are within the same C-

terminal region required for interaction of IpaC with intermediate filaments [7], raising the possibility that IpaC either interacts with multiple host proteins simultaneously or undergoes a series of conformational changes that enable sequential interactions with different host proteins. The latter would potentially allow for distinct signals to direct docking and, once the docking process is complete, activation of effector secretion. Moreover, in other systems, Src phosphorylation of vimentin leads to depolymerization of the vimentin filaments [39], raising the possibility that Src recruitment to the site of *Shigella* invasion has multiple effects on the host cytoskeletal network. Additional investigations are needed to determine the molecular relationship of docking to activation of effector secretion and to define the molecular mechanisms that coordinate the two processes.

It remains uncertain whether activation of type 3 secretion is generally associated with a conformational change in the translocon pore. Disulfide crosslinking of the *Pseudomonas* translocon pore protein PopD, a homolog of IpaC, blocked effector translocation [10], leading to the authors' interpretation that a conformational change of the pore was necessary to activate secretion. Whereas we were unable to observe an intermediate filament-dependent conformational change to SipC that was similar to the one we observed for IpaC, recent investigations of the *S.* Typhimurium needle suggest that secretion is associated with defined conformational states [16]. This finding suggests that, like *Shigella* and *Pseudomonas* [10], conformational changes in the *Salmonella* pore may also occur, even though they were not observed with the approach we used here. Alternatively, the mechanism of SipC function in docking and secretion activation may be distinct from that of IpaC. The two proteins are similar at the amino acid level, but functional and structural differences are also observed [23, 40, 41]; in particular, SipC may need to attain a more conformationally stable structure within the pore, as *Salmonella* establishes an intracellular niche within a vacuole that is irrelevant for *Shigella*.

Our observation that *S. flexneri* docking to WT cells was blocked by intermolecular crosslinking of IpaC molecules when crosslinking was induced during the initial contact of bacteria with cells but not when crosslinking was induced at 10 minutes after initial contact demonstrates that the processes required for docking are complete within 10 minutes. These data further support a model in which the initial conformation of the translocon pore in the plasma membrane is not competent for docking and that, within 10 minutes, the pore converts to a docking-competent conformation. Our data show that this conformational change is directly induced by the interaction of C-terminal residues of membrane-embedded IpaC with intermediate filaments.

By mapping single cysteine substitutions of IpaC that are amenable to intermolecular disulfide crosslinking, we build on our recent mapping of the topology of natively-delivered IpaC [23]. Disulfide crosslinking requires an oxidizing environment, was induced by the membrane-impermeant oxidant copper chloride (Fig 1), and was blocked by TCEP, a membrane-impermeant reductant (S1 Fig). The successful crosslinking of a subset of IpaC cysteine substitutions using these approaches indicated the presence of the residue in an oxidizing environment accessible from the extracellular side of the plasma membrane, which along with PEG5000-maleimide labeling (Fig 3) confirmed our prior demonstration that the IpaC N-terminal region and the 19 residues closest to the IpaC C-terminus are accessible from the extracellular environment and that the bulk of the IpaC domain C-terminal to the alpha-helix is cytosolic (Figs 1 and 3, and [23]). This orientation of IpaC is similar for docking-competent and docking-incompetent pores (Fig 1 and S2 Fig, and [23]), which indicates that the interaction of IpaC with intermediate filaments causes the repositioning of IpaC residues within the pore in a subtle manner that does not alter the protein's overall orientation in the membrane. Since PEG5000-maleimide is too large to cross the membrane or pass through the pore [7], the

observed accessibility from the extracellular milieu of residues near the IpaC C-terminus to PEG5000-maleimide is consistent with these residues looping back into the pore lumen (Fig 3 and S2 Fig, and [23]). Whereas a possible explanation for the intermediate level of labeling of the C-terminal proximate residues is that a subset of IpaC molecules are present in an alternate conformation, we believe the weaker labeling is most likely due to bound probe sterically hindering access of additional probe molecules to remaining free sulfhydryl groups on IpaC molecules within a given pore. Consistent with this, crosslinking induced by copper, which is significantly smaller than PEG5000-maleimide, was not decreased for residues near the C-terminus relative to residues in the N-terminal domain (Figs 1 and 2).

Our identification of IpaC residues amenable to copper-induced crosslinking demonstrates that for intact membrane-embedded translocon pores, IpaC is packed such that for adjacent molecules, several specific residues lie within 2.05 Å (the length of a disulfide bond) of one another. These data are consistent with the previous finding that the *P. aeruginosa* pore protein PopD, an IpaC homolog, can form copper-induced disulfide crosslinks [10], indicating that individual molecules of PopD are also adjacent to each other within the *P. aeruginosa* pore.

Taken together, our findings are the first demonstration that a host-bacterial protein interaction triggers conformational changes to the bacterial T3SS translocon pore and that these conformational changes enable bacterial docking, which is necessary to activate protein secretion. These findings provide new mechanistic insights into the signals required to activate type 3 secretion.

## Methods

### Bacterial culture

The *S. flexneri* WT strain used in this study is serotype 2a strain 2457T, and all mutants used in this study are isogenic to it. *S. flexneri* were cultured in trypticase soy broth (Becton Dickenson) at 37˚C. The *S.* Typhimurium strain used in this study is SL1344, and all mutants are isogenic to it. *S.* Typhimurium was cultured in L-Broth. For all IpaC and SipC constructs, expression was driven from the pBAD promoter, induced with 1.2% arabinose. The sequences of primers used for PCR are available from the authors upon request. Infections were performed with bacteria grown to exponential phase.

### Cell culture

Vim$^{+/+}$ and Vim$^{-/-}$ MEFs were provided by Victor Faundez (Emory), vimentin is the only intermediate filament expressed in these cells [42]. HeLa (CCL2) cells were obtained from ATCC. All cells were cultured in Dulbecco's Modified Eagles Media (DMEM) supplemented 0.45% glucose and 10% heat-inactivated fetal bovine serum (FBS); they were maintained at 37˚C in humidified air containing 5% $CO_2$. All cells in the laboratory are periodically tested for mycoplasma; cells that test positive are treated or discarded.

### Detection of copper crosslinking

Vim$^{+/+}$ and Vim$^{-/-}$ MEFs were seeded at 3 x 10$^5$ cells per well in a 6-well plate. The cells were washed once with Hank's Balanced Salt Solution (HBSS) containing 4% FBS and 1.2% arabinose. The cells were infected at an MOI of 200 in HBSS containing 4% FBS and 1.2% arabinose, with or without 25 μM copper [10]. Experiments in Fig 1 were performed with copper chloride, which is membrane-impermeant, and experiments in Figs 2 and 4, and S1 Fig were performed with copper phenanthroline, which is membrane-permeable. The bacteria were centrifuged onto cells at 800 *g* for 10 minutes at 25˚C and incubated at 37˚C in humidified air

with 5% $CO_2$ for 10 min. IpaC delivered to cell membranes was recovered as done previously [10]. Briefly, cells were washed with HBSS and lysed with Triton X-100, and bacteria and cellular debris were removed by two successive centrifugations at 21,000 $g$ for 2 minutes each at 25°C.

To test the effect of copper at later stages of infection, experiments were performed as described as above except the infections were in HBSS containing 4% FBS and 1.2% arabinose, and instead of adding copper at the same time as the bacteria, copper phenanthroline was added immediately following the 10-minute centrifugation, to a final concentration of 25 μM.

### Erythrocyte lysis assay

Pore formation in sheep erythrocyte membranes was monitored by assessing the efficiency of erythrocyte lysis [6]. Briefly, $10^8$ erythrocytes were washed with saline, and co-cultured with *S. flexneri* at a multiplicity of 25 bacteria per erythrocyte in 30 mM Tris, pH 7.5. After the bacteria were centrifuged onto the erythrocytes at 2,000 $g$ for 10 minutes at 25°C, bacteria and erythrocytes were co-cultured for 30 minutes at 37°C in humidified air with 5% $CO_2$. Bacteria and erythrocytes were then mixed by pipetting and again centrifuged at 2,000 $g$ for 10 minutes at 25°C. As a control for lysis, a portion of uninfected erythrocytes were treated with 0.02% SDS. The supernatants were collected, and the abundance of hemoglobin released was determined spectrophotometrically by determining $A_{570}$ using a Wallac 1,420 Victor$^2$ (Perkin Elmer). For experiments that included copper, copper phenanthroline was added with the bacteria at 25 μM.

### Translocation and docking

HeLa cells or Vim$^{+/+}$ and Vim$^{-/-}$ MEFs were seeded at 3 x $10^5$ cells per well on coverslips in a 6-well plate. The cells were infected with *S. flexneri* [43, 44] harboring the TSAR reporter [37] at a multiplicity of infection (MOI) of 200 in the presence or absence of 25 μM copper phenanthroline. The TSAR reporter expresses GFP when the type 3 effector OspD is secreted through the T3SS [37], thereby serving as a reporter of T3SS effector secretion. Bacteria were centrifuged onto cells at 800 $g$ for 10 minutes at 25°C. The infection was carried out for an additional 40 minutes at 37°C in humidified air with 5% $CO_2$. Cells were washed three times with PBS, fixed with 3.7% paraformaldehyde for 20 minutes at 25°C, and washed with PBS, and DNA was stained with Hoechst. Bacteria in which T3SS secretion was activated produce GFP [37], and to facilitate quantification of bacteria by fluorescent microscopy, all bacteria express mCherry. For experiments testing the effects of copper, nine random microscopic images were collected per coverslip, and the data were averaged.

### Labeling of cysteines with PEG5000-maleimide

To test the accessibility of IpaC or SipC residues from the extracellular milieu during infection, HeLa cells or Vim$^{+/+}$ and Vim$^{-/-}$ MEFs were seeded in 6-well plates. A total of 8 x $10^5$ HeLa cells or 4.8 x $10^6$ MEFs were used for each strain tested. Cells were washed once with 50 mM Tris, pH 7.4, supplemented with 150 mM NaCl and 1.2% arabinose. For infections of HeLa cells, to enhance the efficiency of translocon pore insertion into the plasma membrane, all bacteria expressed the *E. coli* adhesin Afa-1 [7, 45]. Cells were infected at a MOI of 500 in 50 mM Tris, pH 7.4, supplemented with 150 mM NaCl, 1.2% arabinose, and 2.5 mM PEG5000-maleimide. Bacteria were centrifuged onto the cells at 800 $g$ for 10 minutes at 25°C and incubated at 37°C in humidified air with 5% $CO_2$ for 20 minutes. Membrane-enriched fractions containing IpaC were isolated, as done previously [7, 8]. Briefly, the cells were washed three times with ice-cold 50 mM Tris, pH 7.4, and scraped in 50 mM Tris, pH 7.4, containing protease

inhibitors (Protease inhibitor cocktail, complete mini-EDTA free, Roche). Scraped cells were pelleted at 3,000 $g$ for 3 minutes at 25˚C, resuspended in 50 mM Tris, pH 7.4, containing protease inhibitors and 0.2% saponin, and incubated on ice for 20 minutes. The suspension was pelleted at 21,000 $g$ for 30 minutes at 4˚C. The supernatant, which contains the cytosolic fraction, was decanted into a fresh tube. The pellet was resuspended in 50 mM Tris, pH 7.4, containing protease inhibitors and 0.5% Triton X-100, incubated on ice for 30 minutes, and pelleted at 21,000 $g$ for 15 minutes at 4˚C. The supernatant from this spin consists of the membrane-enriched fraction, and the resulting pellet consists of the detergent-insoluble fraction, which includes intact bacteria. The efficiency of PEG5000-maleimide labeling was monitored by assessing the gel shift of IpaC or SipC by western blot. The following antibodies were used for western blots: rabbit anti-IpaC (gift from Wendy Picking; diluted 1:10,000), rabbit anti-GroEL (catalog no. G6352; Sigma) (1:1,000,000), rabbit anti-caveolin-1 (catalog no. C4490; Sigma) (1:1,000), mouse anti-SipC (gift from Jorge Galaán; 1:10,000), goat anti-rabbit conjugated with horseradish peroxidase (HRP) (Jackson ImmunoResearch, catalog no. 115-035-003, 1:5,000), and goat anti-mouse conjugated with HRP (catalog no. 111-035-003; Jackson ImmunoResearch) (1:5,000).

### Microscopy and image analysis

Microscopic images were collected using either a Nikon TE-300 or a Nikon TE2000-S microscope equipped with Chroma Technology filters, a Q-Imaging Exi Blue camera (Q-Imaging), and I-Vision software (BioVision Technologies). Unless otherwise noted, images were collected in a random manner across the coverslip. Images were assembled and digitally pseudo-colored using Photoshop (Adobe) software.

For band densitometry of western blots, following chemiluminescent detection of signals by film, the film was scanned using an Epson Perfection 4990 scanner, and band intensity was determined using ImageJ (NIH).

### Statistical analysis

Unless otherwise noted, all data presented were collected from at least three independent experiments that were conducted on independent days using independent cultures. Unless otherwise noted, one replicate was performed per independent experiment. Individual data points presented in graphs represent independent experimental measurements; if more than one replicate was performed within an experiment, the data point depicted is the mean of the dependent replicates. All statistical analysis was performed using GraphPad Prism (GraphPad Software) or Excel. For comparison of two groups, to determine whether means were statistically different, a Student's t-test was used. Unless otherwise noted, for comparison of three or more groups, to determine whether means were statistically different among groups, a one-way ANOVA was performed followed by a Dunnett's *post hoc* test, or a two-way ANOVA was performed followed by a Sidak *post hoc* test.

### Supporting information

**S1 Fig. Identification of single cysteine substitutions in IpaC that support intermolecular disulfide bond formation during infection. (a-b)** Copper does not inhibit the formation of translocon pores. Erythrocyte lysis as a result of pore formation in cell membranes during co-culture of erythrocytes with *S. flexneri*. *S. flexneri* Δ*ipaC* strains producing cysteine substitution derivatives of IpaC in the presence or absence of copper. Representative image of released hemoglobin in the supernatants of co-cultured erythrocytes (**a**). Efficiency of erythrocyte lysis, as a function of the abundance of hemoglobin in the co-culture supernatants, quantified by

$A_{570}$ (**b**). (**c-d**) Gel shift of IpaC dimer bands following exposure to copper is diminished by addition of the membrane-impermeant reductant TCEP. Representative western blot of IpaC (**c**). Densitometry of the slow migrating bands (**d**). Graphed data are presented as mean ± SEM of three or more independent experiments. \*, $p<0.05$; \*\*, $p<0.01$; N.S., not significant. One-way ANOVA with Dunnett's *post hoc* test (**b**). Student's t-test (**d**).
(TIF)

**S2 Fig. Orientation in the plasma membrane of IpaC R362W, an IpaC derivative that does not interact with intermediate filaments.** Accessibility of membrane-embedded IpaC R362W to labeling with PEG5000-maleimide upon infection of HeLa cells with *S. flexneri* producing the indicated single cysteine substitution derivatives of IpaC R362W. (**a**) Gel shift of PEG5000-maleimide labeled IpaC in the plasma membrane-enriched fraction of infected HeLa cells. Representative western blot of IpaC. IpaC-PEG5000, IpaC R362W derivatives labeled with PEG5000-maleimide; IpaC, unlabeled IpaC R362W derivatives; caveolin-1, plasma membrane protein; GroEL, bacterial cytosolic protein. (**b**) Relative accessibility of IpaC R362W cysteine substitutions. Densitometry analysis of IpaC-PEG5000 bands from experiments represented in panel **a**. Two independent experiments; mean ± SEM. (**c-d**) Efficiency of pore formation in mammalian membranes as measured by erythrocyte lysis during co-culture of erythrocytes with *S. flexneri* Δ*ipaC* strains producing the indicated cysteine substitution derivatives of IpaC R362W. (**c**) Representative images of hemoglobin released into the supernatants of co-cultured erythrocytes. (**d**) Efficiency of erythrocyte lysis, as a function of the abundance of hemoglobin in the co-culture supernatants, quantified by $A_{570}$ in experiments represented in panel **c**. Three independent experiments for each cysteine mutant; mean ± SEM. Strains producing an IpaC R362W cysteine substitution were not statistically different from the strain producing IpaC R362W. \*\*\*, $p<0.001$. Two-way ANOVA with a Dunnett's *post hoc* test. (**e**) IpaC R362W is inserted in mammalian membranes at an efficiency similar to that of WT IpaC. The abundance of WT IpaC and IpaC R362W in the membrane-enriched fractions of Vim$^{+/+}$ MEFs. Mean ± SEM from three independent experiments. No significant difference between means (Student's t-test).
(TIF)

**S3 Fig. Independent experimental replicates for data presented in Fig 3e and 3f.** PEG5000--maleimide labeling of sulfhydryl groups in cysteine substitution derivatives in the context of WT IpaC or IpaC R362W during *S. flexneri* infection of HeLa cells. Western blots from each of six independent experiments performed.
(TIF)

**S4 Fig. Accessibility of residues of the *S.* Typhimurium translocon pore protein SipC from the extracellular surface are not altered by the presence of intermediate filaments.** PEG5000-maleimide labeling of sulfhydryl groups in cysteine substitution derivatives in the context of WT SipC during *S.* Typhimirium infection of Vim$^{+/+}$ and Vim$^{-/-}$ cells. (**a-d**) Representative western blots of PEG5000-maleimide labeling of SipC cysteine derivates in membrane-enriched fractions from Vim$^{+/+}$ (**a**) and Vim$^{-/-}$ (**c**) MEFs. Densitometry analysis of bands corresponding to SipC-PEG5000 from (**a**) and (**c**), respectively (**b** and **d**). (**e-f**) Direct comparison of PEG5000-maleimide labeling of SipC upon infection of Vim$^{+/+}$ and Vim$^{-/-}$ MEFs with *S.* Typhimurium Δ*sipC* producing S18C or A38C. (**e**) Representative western blots. (**f**) Densitometry analysis of bands corresponding to SipC-PEG5000 from (**e**). SipC-PEG, SipC derivatives labeled with PEG5000-maleimide; SipC, unlabeled SipC derivatives; Caveolin-1, marker of eukaryotic plasma membrane; GroEL, bacterial cytosolic protein. Graphed data are presented as mean ± SEM of two (**c-d**) or three (**a-b** and **e-f**) independent experiments. N.S.,

not significant. Two-way ANOVA with Sidak *post hoc* test.
(TIF)

## Acknowledgments

We thank D. Borden Lacy, Daniel Kahne, and members of the laboratories of Marcia Goldberg, Cammie Lesser, and Amy Barczak for helpful discussion. We thank Poyin Chen for critical reading of the manuscript. We thank Wendy Picking, Claude Parsot, and Jorge Galán for reagents.

## Author Contributions

**Conceptualization:** Brian C. Russo, Marcia B. Goldberg.

**Data curation:** Brian C. Russo, Jeffrey K. Duncan.

**Formal analysis:** Brian C. Russo, Jeffrey K. Duncan, Marcia B. Goldberg.

**Funding acquisition:** Brian C. Russo, Marcia B. Goldberg.

**Investigation:** Brian C. Russo, Jeffrey K. Duncan, Alexandra L. Wiscovitch, Austin C. Hachey.

**Methodology:** Brian C. Russo, Jeffrey K. Duncan, Alexandra L. Wiscovitch, Austin C. Hachey.

**Project administration:** Marcia B. Goldberg.

**Resources:** Brian C. Russo, Marcia B. Goldberg.

**Supervision:** Marcia B. Goldberg.

**Writing – original draft:** Brian C. Russo.

**Writing – review & editing:** Jeffrey K. Duncan, Marcia B. Goldberg.

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
