## [Decision Letter · Decision Letter 0]

23 Jul 2019

Dear Dr. Goldberg,

Thank you very much for submitting your manuscript "Activation of Shigella flexneri type 3 secretion requires a host-induced conformational change to the translocon pore" (PPATHOGENS-D-19-01090) for review by PLOS Pathogens. Your manuscript was fully evaluated at the editorial level and by independent peer reviewers. The reviewers appreciated the attention to an important problem, but raised some substantial concerns about the manuscript as it currently stands. These issues must be addressed before we would be willing to consider a revised version of your study. We cannot, of course, promise publication at that time.

We therefore ask you to modify the manuscript according to the review recommendations before we can consider your manuscript for acceptance. Your revisions should address the specific points made by each reviewer.

(1) A letter containing a detailed list of your responses to the review comments and a description of the changes you have made in the manuscript. Please note while forming your response, if your article is accepted, you may have the opportunity to make the peer review history publicly available. The record will include editor decision letters (with reviews) and your responses to reviewer comments. If eligible, we will contact you to opt in or out.

(2) Two versions of the manuscript: one with either highlights or tracked changes denoting where the text has been changed; the other a clean version (uploaded as the manuscript file).

Additionally, to enhance the reproducibility of your results, PLOS recommends that you deposit your laboratory protocols in protocols.io, where a protocol can be assigned its own identifier (DOI) such that it can be cited independently in the future. For instructions see http://journals.plos.org/plospathogens/s/submission-guidelines#loc-materials-and-methods

We hope to receive your revised manuscript within 60 days. If you anticipate any delay in its return, we ask that you let us know the expected resubmission date by replying to this email. Revised manuscripts received beyond 60 days may require evaluation and peer review similar to that applied to newly submitted manuscripts.

Sincerely,

Michael Kolbe

Guest Editor

PLOS Pathogens

Guy Tran Van Nhieu

Section Editor

PLOS Pathogens

Kasturi Haldar

Editor-in-Chief

PLOS Pathogens

orcid.org/0000-0001-5065-158X

Grant McFadden

Editor-in-Chief

PLOS Pathogens

orcid.org/0000-0002-2556-3526

All reviewers got consensus that this work should advance our understanding of the role of IpaC during host cell invasion.

Reviewers raised the question whether the conformational change found in IpaC is a general mechanism of T3SS-host interaction. It would be nice if you could test if translocons formed by IpaC-orthologuous (from Pseudomonas or Salmonella) do also change their conformation upon host contact.

Reviewer's Responses to Questions

**Part I - Summary**

Reviewer #1: The presented experiments are well designed and well conducted, controls thoughtfully planned and extensively performed, tools validated in previous papers, experiments allow clear cut conclusion. The authors use MEF cells which is a logical choice, since they found several IFs to be involved in IpaC interaction (Russo et al 16), and in MEFs vimentin Kos are available.

It would have been nice if authors would have explored a bit more the open directions. This could have been either (i) on a more mechanistical level with a biophysical approach, or (ii) to explore general meaning in host-pathogen interactions of T3SSs with host IFs in comparison with other pathogens.

- Authors did not generate new tools, all was published and used already by them or others.

- With regard to research: Several teams hypothesized that a conformational change initiates secretion. It has been shown already by other team that the T3SS IpaC homolog PopD undergoes conformational change required for secretion. It has been unknown what triggers secretion. The Goldberg team previously showed that IpaC-IF interactions after translocon formation is required for stable docking and thus continuing secretion. They proposed this interaction induces conformational change. Here they show that the trigger is their described IF-IpaC interaction with a singular experimental strategy.

I would have appreciated if the authors developed this further, to verify this with other techniques. Or to explore further the mechanistic consequences, compare different T3SS of different pathogens etc.

Reviewer #2: The manuscript by Russo et al. presents one of the most detailed views to date of docking between the type III secretion needle tip, and the translocation pore. The authors use disulfide bonds introduced into one of the pore-forming translocator proteins, IpaC, to demonstrate that tethering of translocator proteins can specifically interfere with this docking step. Moreover, the authors demonstrate by assaying accessibility of cysteine residues engineered into IpaC, that binding of IpaC to intermediate filaments, a critical step for docking, induces a conformational change. Taken together, these data indicate that binding of IpaC to intermediate filament proteins induces a conformational change in the translocation pore, which then facilitates docking of the needle-tip to the pore. The authors also present preliminary data that argues that triggering of effector secretion involves a discrete conformational step that follows docking, but the data for this are weaker. Taken together, the work presented here represents a significant step forward in our understanding of translocon function, which is a critical component of this important virulence factor.

Reviewer #3: General comments

The manuscript builds on previous work from the group on IpaC, a component of the translocon of the T3SS of Shigella. The authors have introduced cysteine residues into the protein and examined their ability to be cross linked in the presence/absence of copper to act as an oxidant. The effect of the membrane-impermeable reductant TCEP was also assessed. The work relies on changing residues in a protein for which folding is critical, in combination with calcium/TCEP to infer changes in accessibility of residues, conformation, and thence infer these changes could lead to changes in secretion by transducing a signal via a structural alteration down the T3SS needle to the cytoplasm where secretion hierarchy is determined.

The work is well written, and results clearly presented. I have provisos about the overall approaches, and how much weight can be placed on the results, but IpaC and related proteins are difficult to work with at a structural level, so the approaches taken by the authors are reasonable given the challenges of understanding the how the translocon works at the atomic level. However could they confirm their findings using other chemicals with similar or mismatched properties to Ca and TCEP?

**Part II – Major Issues: Key Experiments Required for Acceptance**

Reviewer #1: 1. Is the IF-IpaC conformational change a general T3SS-host interaction mechanism for effector secretion?

A. Comparison with P. aeruginosa: Are IFs like vimentin involved in the conformational changes described for the IpaC-homolog? Infect their Vim -/- cell line with P. aeruginosa and the decribed PopD mutants (is it possible that this is the missing triggering factor in the 2016-P. aeruginosa-publication the IF-PopD interaction?)

B. Or: Comparison with Salmonella: Both IpaC and SipC get activated via IFs (Russo…Goldberg 2016). In Russo et al 2019 they showed that the IpaC equivalent positions at the N-terminus of SipC are extracellular and accessible for modification; they have established SipC mutants (S18C, A38C). Here, they showed that in Shigella these mutants are crosslinked and useful tool to monitor conformational IpaC changes during docking and injection (Fig. 2, 3, 4). Can they show a conformational, vimentin-dependent change during Salmonella T3SS injection? What are differences?

C. They previously found orientation of the last C-terminal aa to lie inside the pore accessible from the extracelluar space, but not for Salmonella (Russo et al 2019). What happens when the last aa of the C-terminus of Salmonella SipC is changed to the IpaC sequence? Is there also a conformational change that involves SipC-IF interaction?

2. The C-terminus has been implied in actin remodeling via Src kinase, exactly the same region that they found to be involved in vimentin binding. Src phosphorylation of vimentin has been described that leads to vimentin disassembly (Oncogene 2019). Can they discuss this with regard to their model?

Reviewer #2: Line 287 last section re: Fig. 4, the decrease in TSAR for A38C is from 80% to 60% in 2 independent experiments. The defect in TSAR is minor, and two biological replicates is too few to really make a strong statement here. Couldn’t any short delay in docking account for this defect? After all, we don’t know if partial oxidation of IpaC in the pore prevents or delays docking. The fundamental question, does triggering of effector secretion derive from docking (an “induced fit” type model of triggering), or does it require a secondary conformational change, is interesting. The Fig. 4 data may argue toward the latter, but I don’t think the data are strong enough to make that call. I think the competing models and the data should be discussed, with some caveats, in the discussion, since this is a useful conversation for the field.

If the authors leave the Fig. 4 data in, then I think they will need an additional biological replicate for the A38C TSAR data (Fig. 4C), since this is the crux of the figure, and 2 seems too few, particularly with the modest effect of adding the copper after spinning down the bacteria.

Reviewer #3: Line 189, conclusion correct? The reduction in docking in Vim-/- cells is substantial, and it is not certain whether they can conclude that calcium is having no effect, and there is very little docking in the first place.

3E/F is a critical experiment, and quantification is not obvious from the blots provided. The authors should include all blots that were used to generate these data.

**Part III – Minor Issues: Editorial and Data Presentation Modifications**

Reviewer #1: • Fig 1 as supplementary figure that only shows that crosslinking of cysteine substitution mutants (that have been already described in previous paper Russo et al 2019 to be accessible and able to modify) actually works. Quantification in (d) contains mutants L117C and S117C, where neither the WB are shown in (c), nor are they mentioned at all in the text. Since authors explain that only extracellular accessible residues can be modified (which they discovered already in Russo et al 2019), they show non-crosslinked, cytosolic residues again in Fig. 1 that are neither further investigated, nor discussed in the paper (S101C, S103C, S108C, A119C, F136C, A313C). In addition, authors perform in Fig. 3 experiments with modified residues 271, 349, and 358, but do also not include them here in their crosslinking control. Since they always do controls in WBs of single experiments, actually this figure is not needed anyway.

• Fig. 1b: Nice scheme that helps with domain organization. Indicating the position of crucial mutants could be helpful for reader to directly localize them in protein context.

• Put a model in suppl fig. in main text. Probably needs revision with indicating single steps that could resemble conformational changes. Docking competent one: extracellularly accessible loop would need to go and bind Vim to open pore, it should block pore if inside pore?

• Figure 4 could be merged with Fig. 3

• Findings could be discussed more based on literature (25 references)

• Paper could be written more concise

Reviewer #2: Line 163: “Thus, the formation of disulfide bonds at S17C, A38C, S63C, A106C, K350C, A353C, and likely some of the bonds formed at A363C, are accessible from the extracellular environment.” There is a problem with these experiments. The authors don’t know what percentage of IpaC detected in these experiments is actually part of an assembled translocation pore. Translocator proteins are exported at a low rate (leakage) before host cell contact, so IpaC detected in these experiments could be stuck to the outside and not properly inserted. They could also be inserted, but not part of a translocation pore. So, to my mind, the disulfide bond clearly occurs in the pore if there is an associated phenotype that is DSB-dependent (e.g. A38C and A353C), but the converse is less clear.

Line 399 onward: Not sure what this is about. From the cited paper: “Given that PopDBpyFL efficiently interacts with PopB, the lack of increase in anisotropy was indicative of the presence of at least two or more PopDBpyFL subunits per PopB/PopD hetero-complex. Taken together, these results showed that PopD assembles homo-oligomeric structures in membranes, but when added together with PopB it forms hetero-oligomeric structures of yet undetermined size and stoichiometry.” I don’ think the data in the cited paper demonstrates an exact alternating PopB-PopD pattern, and therefore don’t conflict with the data from cell-based assays. Although the caveat noted by the authors, that in vitro systems can produce data that do not match data found cell-based systems, is valid.

What the authors fail to mention is that in ref. 9, the authors demonstrate that using a disulfide bond to trap the PopD homodimer specifically interferes with host-cell sensing. The interpretation is that the pore has to undergo a conformational change to trigger effector secretion. This is obviously reminiscent of the model proposed here (and was derived by a similar methodology). I think this should be mentioned in the discussion (and perhaps the intro, line 86, could be modified to set this up). On the whole, the data support the findings in this manuscript, using a different bacterium, so this is a good thing.

Fig. 4a. Looks like more than 1bug/cell, so the images don’t really match up well with the quantitation data. Maybe a different set of images could illustrate this point better. Or is the average of 1bug/cell derived from 10 bugs attaching to 1, and 0 to 9 cells?

Line 288 “Since we found that intermolecular crosslinking of at the time …”, I would remove the “of”

Line 475 should “adhesion” be “adhesin”?

Reviewer #3: Minor comments

Line 288 : re-write sentence

Line 351, there is some evidence for this, and some against. Include references here.

PLOS authors have the option to publish the peer review history of their article (what does this mean?). If published, this will include your full peer review and any attached files.

Reviewer #1: No

Reviewer #2: No

Reviewer #3: No

---

## [Decision Letter · Decision Letter 1]

30 Sep 2019

Dear Dr. Goldberg,

We are pleased to inform that your manuscript, "Activation of Shigella flexneri type 3 secretion requires a host-induced conformational change to the translocon pore", has been editorially accepted for publication at PLOS Pathogens. 

Before your manuscript can be formally accepted and sent to production, you will need to complete our formatting changes, which you will receive by email within a week. Please note that your manuscript will not be scheduled for publication until you have made the required changes.

IMPORTANT NOTES

(1) Please note, once your paper is accepted, an uncorrected proof of your manuscript will be published online ahead of the final version, unless you’ve already opted out via the online submission form. If, for any reason, you do not want an earlier version of your manuscript published online or are unsure if you have already indicated as such, please let the journal staff know immediately at plospathogens@plos.org.

(2) Copyediting and Proofreading: The corresponding author will receive a typeset proof for review, to ensure errors have not been introduced during production. Please review the PDF proof of your manuscript carefully, as this is the last chance to correct any errors. Please note that major changes, or those which affect the scientific understanding of the work, will likely cause delays to the publication date of your manuscript. 

(3) Appropriate Figure Files: Please remove all name and figure # text from your figure files. Please also take this time to check that your figures are of high resolution, which will improve the readbility of your figures and help expedite your manuscript's publication. Please note that figures must have been originally created at 300dpi or higher. Do not manually increase the resolution of your files. For instructions on how to properly obtain high quality images, please review our Figure Guidelines, with examples at: http://journals.plos.org/plospathogens/s/figures.

(4) Striking Image: Please upload a striking still image to accompany your article if one is available (you can include a new image or an existing one from within your manuscript). Should your paper be accepted, this image will be considered for our monthly issue image and may also appear on our website to feature your article. Please upload this as a separate file, selecting "striking image" as the file type upon upload. Please also include a separate "Other" file with a caption, including credits and any potential copyright information. Please do not include the caption in the main article file. If your image is from someone other than yourself, please ensure that the artist has read and agreed to the terms and conditions of the Creative Commons Attribution License at http://journals.plos.org/plospathogens/s/content-license. Please note that PLOS cannot publish copyrighted images.

(5) Press Release or Related Media: If your institution or institutions have a press office, please notify them about your upcoming paper at this point, to enable them to help maximize its impact. If they will be preparing press materials for this manuscript, please inform our press team in advance at plospathogens@plos.org as soon as possible. We ask that you contact us within one week to plan ahead of our fast Production schedule. If you need to know your paper's publication date for related media purposes, you must coordinate with our press team, and your manuscript will remain under a strict press embargo until the publication date and time. This means an early version of your manuscript will not be published ahead of your final version. 

(6)  PLOS requires an ORCID iD for all corresponding authors on papers submitted after December 6th, 2016. Please ensure that you have an ORCID iD and that it is validated in Editorial Manager.  To do this, go to ‘Update my Information’ (in the upper left-hand corner of the main menu), and click on the Fetch/Validate link next to the ORCID field.  This will take you to the ORCID site and allow you to create a new iD or authenticate a pre-existing iD in Editorial Manager

(7) Update your Profile Information: Now that your manuscript has been provisionally accepted, please log into Editorial Manager and update your profile, if needed. Go to https://www.editorialmanager.com/ppathogens, log in, and click on the "Update My Information" link at the top of the page. Please update your user information to ensure an efficient production and billing process. 

(8) LaTeX users only: Our staff will ask you to upload a TEX file in addition to the PDF before the paper can be sent to typesetting, so please carefully review our Latex Guidelines http://journals.plos.org/plospathogens/s/latex in the meantime.

(9) If you have associated protocols in protocols.io, please ensure that you make them public before publication to guarantee immediate access to the methodological details.

Best regards,

Michael Kolbe

Guest Editor

PLOS Pathogens

Guy Tran Van Nhieu

Section Editor

PLOS Pathogens

Kasturi Haldar

Editor-in-Chief

PLOS Pathogens

orcid.org/0000-0001-5065-158X

Grant McFadden

Editor-in-Chief

PLOS Pathogens

orcid.org/0000-0002-2556-3526

Reviewer Comments (if any, and for reference):

Reviewer's Responses to Questions

**Part I - Summary**

Reviewer #1: I think the authors have done an excellent job to address the reviewers comments (not only ours). I think the mansucript will provide important new insights into T3SS function.

Reviewer #2: The authors did a good job addressing the reviewer comments. I have no further concerns.

Reviewer #3: (No Response)

**Part II – Major Issues: Key Experiments Required for Acceptance**

Reviewer #1: The manuscript is complete.

Reviewer #2: (No Response)

Reviewer #3: (No Response)

**Part III – Minor Issues: Editorial and Data Presentation Modifications**

Reviewer #1: No minor issues

Reviewer #2: (No Response)

Reviewer #3: (No Response)

PLOS authors have the option to publish the peer review history of their article (what does this mean?). If published, this will include your full peer review and any attached files.

Reviewer #1: Yes: Jost Enninga

Reviewer #2: No

Reviewer #3: No

---

## [Editor Report · Acceptance letter]

7 Nov 2019

Dear Dr. Goldberg,

We are delighted to inform you that your manuscript, "Activation of *Shigella flexneri* type 3 secretion requires a host-induced conformational change to the translocon pore," has been formally accepted for publication in PLOS Pathogens.

Best regards,

Kasturi Haldar

Editor-in-Chief

PLOS Pathogens

orcid.org/0000-0001-5065-158X

Grant McFadden

Editor-in-Chief

PLOS Pathogens

orcid.org/0000-0002-2556-3526